# Geographical Patterns of COVID-19 Vaccine Inequality by Race and Ethnicity and Sociodemographic Determinants of Health: Evidence from Louisville, Kentucky

**DOI:** 10.3390/vaccines13121241

**Published:** 2025-12-13

**Authors:** Seyed M. Karimi, Amir Hossein Hassani, Hamid Zarei, Mana Moghadami, Md Yasin Ali Parh, Shaminul H. Shakib, Venetia Aranha, Mohammad Mansouri, Trey Allen, Yuting Chen, Sirajum Munira Khan, Farzaneh Raoofi, Sepideh Poursafargholi, Taylor Ingram, Angela Graham

**Affiliations:** 1Department of Health Management and Systems Sciences, School of Public Health and Information Sciences, University of Louisville, Louisville, KY 40292, USA; hamid.zarei@louisville.edu (H.Z.); mana.moghadami@louisville.edu (M.M.); 2Division of Population Health, Louisville Metro Department of Public Health and Wellness, Louisville, KY 40202, USA; trey.allen@louisvilleky.gov (T.A.); yuting.chen@louisvilleky.gov (Y.C.); angela.graham@louisvilleky.gov (A.G.); 3School of Medicine, Shiraz University of Medical Sciences, Shiraz 71348-51156, Iran; amirhossein.hassani77@gmail.com; 4Department of Biostatistics, University of Michigan, Ann Arbor, MI 48109, USA; yasinp@umich.edu; 5Department of Public Health, College of Health Sciences, Sam Houston State University, Huntsville, TX 77340, USA; shs017@shsu.edu; 6Department of Epidemiology and Population Health, School of Public Health and Information Sciences, University of Louisville, Louisville, KY 40292, USA; venetia.aranha@louisville.edu; 7Department of Economics (DSE), Alma Mater Studiorum Università di Bologna, 40126 Bologna, Italy; mohammad.mansouri3@studio.unibo.it; 8Department of Bioinformatics and Biostatistics, School of Public Health and Information Sciences, University of Louisville, Louisville, KY 40292, USA; sirajummunira.khan@louisville.edu (S.M.K.); sepideh.poursafargholi@louisville.edu (S.P.); 9Department of Mathematics, College of Arts & Sciences, University of Louisville, Louisville, KY 40292, USA; farzaneh.raoofi@louisville.edu; 10Down Syndrome of Louisville, Louisville, KY 40204, USA; taylor.ingram@louisvilleky.gov

**Keywords:** COVID-19 vaccination, vaccine equity, racial and ethnic disparities, social determinants of health, spatial analysis, Jefferson County, Kentucky

## Abstract

Background: Infectious diseases accounted for approximately 18.4% of global mortality in 2019. According to the World Health Organization (WHO), vaccines are available for about 30 potentially lethal diseases. Vaccination prevents substantial mortality and hospitalization. However, its ability to improve overall public health depends on equitable access across all populations, regardless of race, ethnicity, education, or socioeconomic status. Objectives: This study aims to examine how disparities in social determinants of health (SDOH) affect COVID-19 vaccination uptake across Jefferson County, Kentucky. Using ZIP code–level spatial mapping, this study investigates the intersection of SDOH, racial composition, and geographic characteristics to identify inequities and inform equitable interventions. Methods: Data from the Kentucky Immunization Registry (KYIR) were analyzed to assess two-dose COVID-19 vaccination rates at the ZIP code and regional levels in Jefferson County, Kentucky. Vaccination rates were stratified by race and ethnicity and linked with SDOH, including education, employment, insurance status, and income, obtained from the 2021 American Community Survey. Results: By May 2021, vaccination rates ranged from 25.9% in the West region to 57.0% in the Inner East region; by May 2022, these rates increased to 46.2% and 73.9%, respectively. White residents consistently had the highest two-dose vaccination rates (66.4% by May 2022), while Black and Hispanic residents had lower rates (45.7% and 43.9%, respectively). Vaccination rates were strongly correlated with SDOH, especially educational attainment, average family income, and employment rate, underscoring the role of socioeconomic inequities in vaccination disparities. Conclusions: Geographical and racial disparities emphasize the influence of social and economic inequality on vaccine uptake.

## 1. Introduction

Infectious diseases accounted for approximately 18.4% of global mortality in 2019 [1]. Vaccination, a form of active immunization against infectious diseases, is practiced worldwide [2]. According to the World Health Organization (WHO), vaccines are available for about 30 potentially lethal disorders―including measles, mumps, diphtheria, tetanus, human papillomavirus, and, lastly, COVID-19―and can prevent 3.5–5 million deaths annually. For example, widespread vaccination could prevent approximately a quarter of under-five-year mortality in 2019 alone [3]. The importance of national, regional, and global vaccination programs was reiterated with the emergence of the COVID-19 pandemic [1,4]. Racial and ethnic vaccine equality is essential to reducing health disparities and improving overall public health outcomes. Black, Hispanic, and Native American individuals had a higher COVID-19 hospitalization and mortality rate in the U.S. compared to White individuals [5,6]. Despite having a higher risk of COVID-19 hospitalization and mortality due to higher rates of disease exposure and underlying medical conditions in minority groups, COVID-19 vaccination uptake has been lower than average in most racial and ethnic minorities [6,7,8,9].

Understanding the racial and ethnic disparities in the COVID-19 vaccine uptake requires examining their root causes, particularly the role of social determinants of health (SDOH) [10]. Factors such as income, education, housing, and access to healthcare significantly influence individuals’ ability to receive vaccines [11,12]. As race and ethnicity highly correlate with the extent of these barriers in the U.S. [10], they can serve as proxies for SDOH. Particularly, communities of color, primarily Black, Hispanic, and Native American populations, encounter multiple challenges stemming from systemic inequities, historical disinvestment, and socioeconomic vulnerabilities, which substantially impede their access to vaccines [11].

The neighborhood of residence is another key SDOH proxy. Hence, understanding spatial heterogeneity in COVID-19 vaccination rates is essential for developing tailored public health policies [13,14]. Spatial heterogeneity demonstrates that racial and ethnic disparities in vaccination rates are inconsistent and significantly influenced by neighborhood-level SDOH and residential segregation patterns [13,15]. The fact that vaccination rates were lower in communities with higher social vulnerability scores—which often reflect lower socioeconomic status and less access to healthcare—has been highlighted in previous studies, underscoring the potential of social interventions and policy-making to reduce the inequity gap [4,16].

Although significant progress has been made in documenting racial and ethnic disparities in COVID-19 vaccination rates, critical gaps persist in understanding the spatial heterogeneity of these disparities at granular geographic levels. Previous studies have primarily relied on county-level data, which inadequately capture intraregional variations and neighborhood-level dynamics, particularly in areas characterized by high social vulnerability [15,16]. This limitation is particularly evident in some studies that focus on broad county-level analyses, thereby missing the nuances of local disparities within urban areas [17]. Furthermore, the complex interaction between race/ethnicity and residential environment warrants further investigation, as the relationship between vaccine uptake and race/ethnicity may differ across local contexts [18].

Disparities in SDOH in Jefferson County, Kentucky, illustrate the influence of labor force participation, education, and income on COVID-19 vaccination rates. Compared to areas like the Inner East, West Louisville struggles with lower income levels and educational attainment [19]. This study addresses these gaps by utilizing ZIP code-level regional mapping to examine the intersection of SDOH, racial composition, and geographic characteristics, to guide tailored, equitable public health interventions. Although this study focuses on COVID-19 vaccination, it can serve as an example for analyzing immunization data for other infectious diseases, particularly for previously well-controlled diseases such as measles in the U.S. [20].

## 2. Materials and Methods

The dataset for this study was obtained from the Kentucky Immunization Registry (KYIR), provided by the Louisville Metro Department of Public Health and Wellness (LMPHW). The KYIR data includes COVID-19 vaccine recipients’ demographic characteristics, residential ZIP codes, vaccine type, and vaccination dates. The two-dose COVID-19 vaccination rates at the end of the second quarter (31 May 2021) and the sixth quarter (31 May 2022) of the COVID-19 vaccination campaign in Jefferson County, Kentucky, were calculated by race and ethnicity in the county’s ZIP codes and regions. The county includes the City of Louisville, which had a Census 2020 population of 782,969 [21]. Additionally, correlations between ZIP code-level vaccination rates and a set of SDOH (namely, educational attainment, employment status, insurance status, and average family income) were calculated using the Pearson product-moment correlation coefficient. The SDOH measures were obtained from the U.S. Census Bureau’s 2021 American Community Survey (https://data.census.gov/advanced, accessed on 12 August 2024).

The numerator of the calculated vaccination rates was the number of individuals who received at least two doses of a COVID-19 vaccine in each racial and ethnic category, extracted from the KYIR data. The 2021 estimated population in the racial and ethnic categories was used as the denominator of the rates. Single-year ZIP code-level population data by age, race, and ethnicity from the 2010 and 2020 Censuses were used to extrapolate the corresponding 2021 populations.

Racial groups included White, Black, Asian, Multiracial, and Some Other Race. Two racial categories (namely, American Indian/Alaska Native and Native Hawaiian/Other Pacific Islander) have very small populations in the county [21] and were therefore excluded. Ethnicity groups included Hispanic and non-Hispanic.

ZIP codes allocated to P.O. Boxes or to the city’s international airport, and with a population below 1000, were excluded. In addition to the ZIP codes, COVID-19 vaccination rates were calculated for six country regions: West, Southwest, South, Central, Inner East, and Outer East (Figure A1). The regions were formed based on market areas used in the LMPHW in disease and immunization surveillance.

COVID-19 vaccine uptake was examined against a set of sociodemographic determinants of health, namely, the percentage of individuals with various educational levels (reported for populations aged 25 and older), employment status, and average family income at the ZIP code and county region level. Stata v.18 was used to analyze the data, the tables were created in Microsoft Excel, and the maps were created in Python 3.11 (See: https://github.com/Hamidz88/Geographical_Inequality_COVID-19_Vaccine_Race_Ethnicity/tree/main, accessed on 12 August 2024).

## 3. Results

In 2021, the county’s estimated population was 817,446 (Table 1). The Outer East region was the most populous region of the county (242,798; 29.7%), followed by the South (201,031; 24.6%) and Southwest (178,265; 21.8%). The Inner East (74,686; 9.1%) and Central (68,890; 8.4%) had comparatively smaller shares of the population, while the West region (51,776; 6.3%) was the least populous region in the county. The county’s most populous ZIP codes (40214 and 40216) were in the Southwest region and contained about half of the region’s population (Table A1 and Table A2). The racial composition of the county was 63.7% White, 21.0% Black, 8.2% Multiracial, 3.6% Asian, and 3.6% Some Other Race. Also, 7.8% of the population was Hispanic. Black residents of the county had the highest concentration in the West region, where they composed 77.9% of the population. The highest concentration of Asian residents was in the Outer East region (5.5%), and the highest concentration of White residents was in the Inner East and Outer East regions (84.9% and 75.3%, respectively).

Geographical disparities in the two-dose COVID-19 vaccination rate across the county’s regions and ZIP codes were significant and persistent (Figure 1 and Figure 2). On 31 May 2021, two quarters after the start of the vaccination campaign, the rates were the highest in the Inner East and Outer East regions at 57.0% and 50.0%, respectively, and the lowest in the West and Southwest regions at 25.9% and 32.1%, respectively (Table 2). Although the rate of vaccination grew faster in the West and Southwest regions than in other regions after a year―reaching 46.2% and 52.9% on 31 May 2022, respectively―they remained remarkably lower than those in the Inner East and Outer East regions, with rates at 73.9% and 68.0%, respectively (Table 2). The Inner East, West, and Southwest regions’ ZIP codes were largely homogeneous in rate of vaccinations, but there were large disparities within the Outer East region (Figure 1 and Figure 2; Table A3 and Table A4).

Among racial groups, White residents had the highest two-dose vaccination rate at both time points: 47.5% on 5/31/2021 and 66.4% on 31 May 2022 (Figure 1 and Figure 2 and Table 2). Nonetheless, the White vaccination rate significantly varied across county regions and ZIP codes, with Inner East and Outer East regions consistently having the highest rate (for example, 58.4% and 53.4%, respectively, after two quarters of vaccination) and Southwest having the lowest rate (for example, 38.4% after two quarters of vaccination). On the other hand, the county’s Black residents had the lowest two-dose vaccination rate among the racial groups: 26.5% on 31 May 2021 and 45.7% on 31 May 2022. Regional and ZIP code disparities among Black residents’ vaccination rate was substantial as well with a similar pattern to White residents’ rate: Inner East and Outer East regions consistently had the highest rate (for example, 57.5% and 42.5%, respectively, after two quarters of vaccination) and Southwest had the lowest rate (for example, 21.2% after two quarters of vaccination). Substantial geographical disparities in vaccination rates existed for other racial groups as well.

The COVID-19 vaccination rate among the county’s Hispanic residents was the lowest among all racial and ethnic groups: 22.0% on 31 May 2021 and 43.9% on 31 May 2022 (Table 2). Notably, the rate was homogenously low across the county regions―alarmingly low in the West region: 11.2% and 27.9% after two and six quarters of vaccination, respectively.

The largest increases in two-dose COVID-19 vaccine uptake from the end of the second quarter (31 May 2021) to the end of the sixth quarter (31 May 2022) were in the regions with the lowest rates of vaccination: 21.1, 20.8, and 20.3 percentage point increases in the South, Southwest, and West regions, respectively (Table 2). White residents showed a larger increase in vaccination rates than Black residents in these three regions: 21.1, 21.6, and 28.7 percentage-point increases for White residents versus 20.3, 17.8, and 17.9 percentage-point increases for Black residents, respectively.

COVID-19 vaccination rate differences across the county’s geographical units can be primarily explained by their socioeconomic differences, regardless of race or ethnicity. For example, the county’s West region had the lowest vaccination rates (25.9% on 31 May 2021 and 46.2% on 31 May 2022), the lowest average family income ($47,441), the lowest employment rate (57.1%), and the lowest rate of 25-plus population with a bachelor’s or graduate degree (9.2%) (Table 3). On the other hand, the county’s Inner East region had the highest vaccination rates (57.0% on 31 May 2021 and 73.9% on 31 May 2022), the highest average family income ($152,846), the highest employment rate (69.4%), and the highest rate of 25-plus population with a bachelor’s or graduate degree (62.9%). In effect, the ZIP code-level vaccination rate had the largest correlation coefficients with the percentage of the ZIP code population with bachelor’s and graduate degrees (0.90 and 0.85, respectively, on 31 May 2021), among other socioeconomic determinants of health (Table 4).

## 4. Discussion

In this study, the geographic, racial, and ethnic variations in the completion of the COVID-19 vaccine’s two-dose series were evaluated in Jefferson County/City of Louisville, the largest metropolitan area in the State of Kentucky, USA. Significant differences in vaccination uptake were measured across the county’s ZIP codes, both two and six quarters after the start of the COVID-19 mass vaccination campaign. The uptake of the COVID-19 vaccine among individuals of specific races and ethnicities also varied remarkably by ZIP code. The highest overall, not ZIP code-specific, vaccination rates were among the county’s White and Asian residents, while the Black residents had the lowest vaccination rates [9,22,23]. Both the overall and race- and ethnicity-specific rates of the COVID-19 vaccine uptake strongly correlated with SDOH (namely, education level, family income, and employment rate). Particularly, Black residents of predominantly White regions and ZIP codes had a remarkably higher COVID-19 vaccination rate than Black residents of predominantly Black regions and ZIP codes. The result challenges monolithic views of racial vaccine hesitancy. While racial disparities in COVID-19 vaccine uptake are well known, this indicates that spatial heterogeneity at the often-overlooked granular levels, such as ZIP code and census tracts, is much more revealing.

In effect, SDOH are suspected to be the hidden determining factors in racial and ethnic disparities in vaccine uptake. Using county-level COVID-19 vaccination data from the State of Georgia, U.S., a similar study also found lower vaccination uptake among Black residents compared to White residents. The differences were strongly predicted by a social association index that included SDOH such as income inequality and access to healthcare resources [15]. A study performed on data from the City of Milwaukee, Wisconsin, U.S., showed that non-Hispanic Black residents had a significantly lower vaccination rate, both in two-dose completion and in booster shots, than other racial and ethnic groups. The study also showed that the city’s non-Hispanic Black residents had a higher social vulnerability index, calculated based on unemployment rate, family income, and education level, among other factors [24]. Also, a study on the length of time from the first to the second dose of the COVID-19 vaccine showed that Black Americans needed a longer time to complete the COVID-19 vaccine series, but when the estimates were adjusted for SDOH (namely, perceived infection risk and chronic disorders) the time needed to complete the series was not significantly different between Black and White Americans [25]. The findings indicate that race and ethnicity are proxies for SDOH and that there is a need for broader strategies (those addressing disparities in economic opportunities, education, and access to healthcare) to sustainably decrease racial and ethnic disparities in immunization in the U.S. in the long term [26].

In addition to representing racial and ethnic disparities in SDOH, immunization disparities display hesitancy in government-organized pharmaceutical interventions. For example, a multi-state study, using data from five US states (namely, California, Illinois, Ohio, Florida, and Louisiana) collected in the first five months of COVID-19 vaccination, showed more than 50% of Black Americans were hesitant about receiving the COVID-19 vaccine―a hesitancy rate that was approximately 4.66 times greater than that in White residents [27]. Also, a national survey on COVID-19 vaccine hesitancy also showed that Black Americans were more hesitant than White Americans to get vaccinated in the first month of the COVID-19 vaccination campaign in the U.S., although their hesitancy decreased significantly after 6 months, compared with White Americans [28]. A systematic review―investigating the reasons for vaccine hesitancy among Black Americans―found that older age, higher income, Christian beliefs or atheistic views, democratic political views, trust in vaccine efficacy, as well as trust in the healthcare system, medical providers, and non-discriminatory practices had a positive association with higher vaccine acceptance [29]. The precise mechanism—whether it is access or trust—is still a hypothesis that requires testing through further qualitative studies. Hence, strategies to improve racial and ethnic SDOH disparities may need to be combined with policies to strengthen vaccine acceptance by increasing trust in the healthcare system, communicating vaccine efficacy information effectively to historically abused communities, and engaging trusted community stakeholders with extensive networks [30,31,32].

Among the three SDOH examined in this study, education has the highest correlation with the COVID-19 vaccination rate. The percentage of the ZIP code population (25 years and older) with a high school diploma showed a negative correlation with completion of the COVID-19 vaccine series. Subsequently, the percentage of the ZIP code population with a university degree showed a positive correlation with the rate. This correlation weakened from the second to the sixth quarter of vaccination. The significant association between higher educational attainment and vaccine acceptance has been found in other studies as well [33,34]. This finding indicates a correlation between higher education and health literacy and the greater ability of college-educated individuals to interpret data and distinguish information from misinformation. Therefore, health information strategies may prioritize outreach to communities with lower levels of education. Such interventions might include providing easily comprehensible messages with proper illustrations.

Family income had the second-highest correlation with the COVID-19 vaccination rate in this study: the higher the average family income in a ZIP code, the higher the completion rate of the COVID-19 vaccination series in that ZIP code. This finding is consistent with the literature [35]. Employment rate had the third-highest correlation with the COVID-19 vaccination rate in this study: the higher the employment rate in a ZIP code, the higher the completion rate of the COVID-19 vaccination series in that ZIP code. This finding might be explained by the fact that individuals who needed to attend their workplace were more at risk of contracting COVID-19 and accepted vaccination more readily than their unemployed counterparts. Nonetheless, workplace behaviors and vaccine acceptance share a bidirectional relationship when fully vaccinated individuals are not experiencing breakthrough infections: a higher need for interpersonal relationships increases vaccination uptake, and vaccination enables the relaxation of social distancing protocols [36]. Promoting vaccination among individuals with and without active employment requires specific considerations and may be adjusted by income and education.

While this study provides insight into how socioeconomic and spatial inequities influence vaccination uptake, it has several limitations. First, the data used in this study contained incomplete demographic information due to reporting or data-entry errors. To resolve the problem, missing demographic information was imputed using the respective demographic information reported for other vaccination doses. Another major challenge was the inconsistent reporting of Multiracial and Some Other Race, leading to percentages for these two groups exceeding the total census population. Such inconsistencies were addressed by redistributing the excess number of Multiracial and Some Other Race individuals among other races, according to an algorithm described in a prequel article [22]. Moreover, individual-level behavior cannot be perfectly inferred from the findings of this study, which was conducted at the ZIP code level and thus overlooks individual-level variation within ZIP codes, as residents in the same ZIP code might not share the same socioeconomic conditions or access to vaccines. For example, high-income individuals in low-income ZIP codes (or vice versa) act as outliers. Additionally, individual may have moved their residential ZIP code between the 2020 SDOH data collection and the 2021–2022 vaccine data collection. Hence, the accuracy of the connection between vaccination data and SDOH factors may be impacted by this disparity. Finally, factors such as vaccine hesitancy, access to healthcare facilities, and public health outreach initiatives were beyond the scope of this study, limiting the ability to establish causal relationships between SDOH and vaccination disparities.

## 5. Conclusions

This study addresses the geographical, racial, and cultural variations in COVID-19 immunization rates across Jefferson County. The study shows that vaccination rates are lower in areas with greater economic challenges, especially among minority communities. Black and Hispanic populations in ZIP codes with higher levels of vulnerability, like lower income, less education, and higher unemployment, observed lower vaccination uptake. These findings underscore the need for public health efforts to increase immunization in underserved areas. Some efforts to do this include improving access to vaccines through mobile units and local clinics, partnering with trusted community organizations and leaders, and launching culturally relevant outreach programs. It is also important to continue tracking vaccination trends at the neighborhood level, as this helps ensure resources are allocated effectively and allows for measurement of progress toward more equitable health outcomes. Prioritizing these efforts is essential, not only to reduce existing vaccination gaps but also to build trust in marginalized communities, so they are better prepared for future public health challenges.

## Figures and Tables

**Figure 1 vaccines-13-01241-f001:**
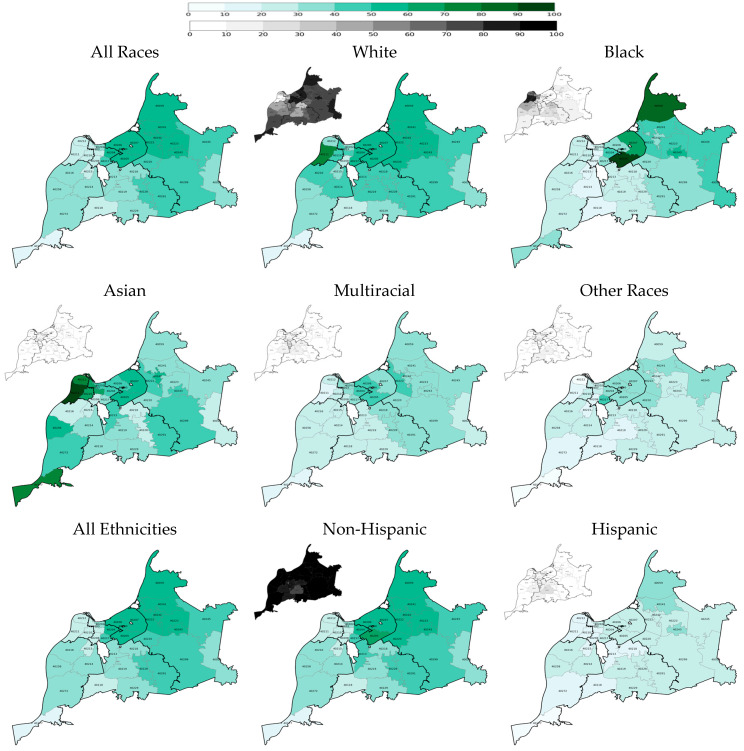
Two-dose COVID-19 vaccination rates at the end of the second quarter of the vaccination campaign on 31 May 2021 by race, ethnicity, and ZIP code (population density maps located at the top left corners).

**Figure 2 vaccines-13-01241-f002:**
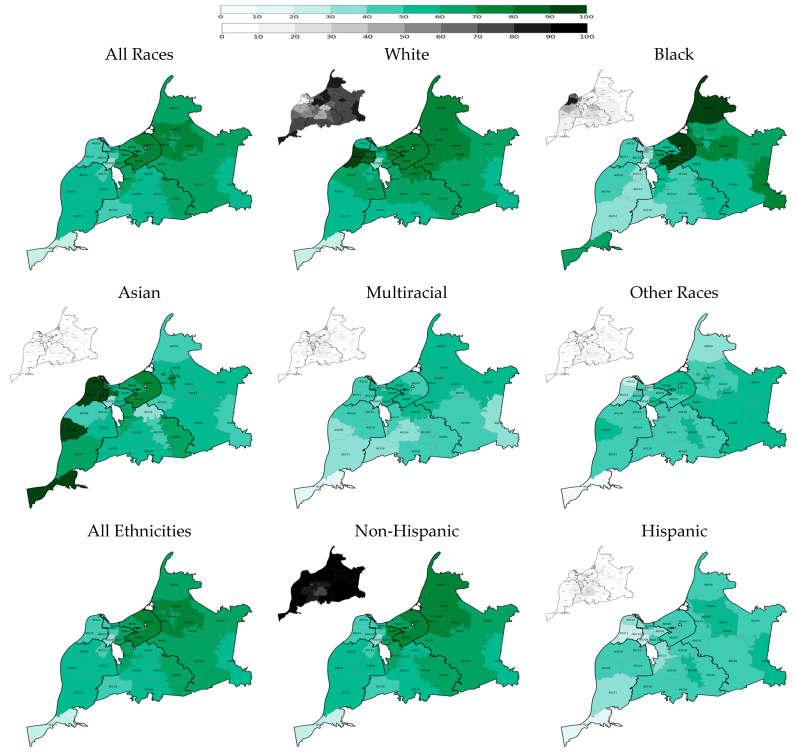
Two-dose COVID-19 vaccination rates at the end of the second quarter of the vaccination campaign on 31 May 2022 by race, ethnicity, and ZIP code (population density maps located at the top left corners).

**Table 1 vaccines-13-01241-t001:** Population of Jefferson County, Kentucky, in 2021 by county region and race.

		Race						
County Region	Overall	White	Black	Asian	Multi-Racial	Some Other Races	Ethnicity	
							Non-Hispanic	Hispanic
Population:								
Overall	817,446	520,643	171,549	29,258	66,753	29,243	751,961	63,842
West	51,776	8010	40,325	109	2569	763	50,626	1200
Southwest	178,265	103,495	45,748	5279	15,926	7817	161,562	16,243
South	201,031	120,815	38,635	6998	21,644	12,939	172,839	27,371
Central	68,890	42,137	19,096	1812	4485	1360	65,995	3045
Inner East	74,686	63,383	3978	1602	4546	1177	71,609	3017
Outer East	242,798	182,803	23,767	13,458	17,583	5187	229,330	12,966
Shares:								
	Vertical %	Horizontal %						
Overall	-	63.7	21.0	3.6	8.2	3.6	92.0	7.8
West	6.3	15.5	77.9	0.2	5.0	1.5	97.7	2.3
Southwest	21.8	58.1	25.7	3.0	8.9	4.4	90.9	9.1
South	24.6	60.1	19.2	3.5	10.8	6.4	86.3	13.7
Central	8.4	61.2	27.7	2.6	6.5	2.0	95.6	4.4
Inner East	9.1	84.9	5.3	2.1	6.1	1.6	96.0	4.0
Outer East	29.7	75.3	9.8	5.5	7.2	2.1	94.6	5.4

Notes: Two races (American Indians and Alaska Natives, also Native Hawaiian or Other Pacific Islanders) are not included in the population estimates due to their small size in the county. Source: Authors’ estimates using ZIP code-level population data by race from the Censuses 2010 and 2020.

**Table 2 vaccines-13-01241-t002:** Two-dose COVID-19 vaccination rates at the end of the second and sixth quarters of the vaccination campaign in Jefferson County, Kentucky, on 31 May 2021 and 31 May 2022, respectively, by race, ethnicity, and ZIP code.

			Races						
Assessment Date	County Region	Overall	White	Black	Asian	Multi-Racial	Some Other Races	Ethnicities	
								Non-Hispanic	Hispanics
Second Quarter (Q2) 31 May 2021	Overall	40.6	47.5	26.5	36.5	30.0	23.4	42.1	22.0
	West	25.9	41.7	23.1	85.0	21.1	14.9	26.3	11.2
	Southwest	32.1	38.4	21.2	32.9	24.7	21.0	33.2	19.8
	South	35.8	41.9	25.4	29.9	27.5	21.3	38.1	20.8
	Central	40.3	47.2	22.9	39.0	33.5	26.9	41.1	22.5
	Inner East	57.0	58.4	57.5	53.0	41.0	34.9	58.2	28.0
	Outer East	50.0	53.4	42.8	38.4	35.6	30.2	51.3	27.2
Sixth Quarter (Q6) 31 May 2022	Overall	59.8	66.4	45.7	52.3	46.1	46.7	61.1	43.9
	West	46.2	70.4	41.0	100.0	45.4	37.8	46.6	27.9
	Southwest	52.9	60.0	39.0	55.1	41.9	45.5	53.6	44.5
	South	56.9	63.0	45.7	43.6	43.6	47.2	58.7	44.0
	Central	55.5	61.3	40.9	52.1	47.8	39.8	56.4	35.4
	Inner East	73.9	74.9	83.6	71.6	51.9	48.0	75.1	44.2
	Outer East	68.0	70.8	65.3	53.0	51.6	50.5	69.2	47.0
Change from Q2 to Q6	Overall	19.2	18.9	19.2	15.8	16.1	23.3	19.0	21.9
	West	20.3	28.7	17.9	15.0	24.3	22.9	20.3	16.7
	Southwest	20.8	21.6	17.8	22.2	17.2	24.5	20.4	24.7
	South	21.1	21.1	20.3	13.7	16.1	25.9	20.6	23.2
	Central	15.2	14.1	18.0	13.1	14.3	12.9	15.3	12.9
	Inner East	16.9	16.5	26.1	18.6	10.9	13.1	16.9	16.2
	Outer East	18.0	17.4	22.5	14.6	16.0	20.3	17.9	19.8

Notes: The intensity of green in the table indicates vaccination rate: the higher the vaccination rate, the bolder the green. The intensity of brown indicates the degree of increase in the vaccination rate from 31 May 2021 (Q1) to 31 May 2022 (Q6): the greater the increase, the bolder the brown.

**Table 3 vaccines-13-01241-t003:** A set of social determinants of health for Jefferson County, Kentucky, by county region.

	% with	% with	% with		Average
County Regions	High School Degree	Bachelor Degree	Graduate Degree	% Employed	Family Income
Overall	26.6	20.7	14.1	65.9	$107,580.6
West	38.1	6.2	3.0	57.1	$47,441.9
Southwest	37.3	11.2	5.3	63.9	$74,088.4
South	33.3	15.5	7.9	69.2	$87,802.1
Central	25.5	20.6	15.8	62.3	$85,107.3
Inner East	12.3	33.4	29.5	69.4	$152,846.2
Outer East	16.4	30.2	21.8	66.5	$148,487.7

Sources: ZIP code level 5-year estimates from the American Community Survey 2021 were used to calculate regional values. ZIP code population shares were used as weights.

**Table 4 vaccines-13-01241-t004:** The ZIP code-level correlation between the two-dose COVID-19 vaccination rate and a number of sociodemographic determinants of health in Jefferson County, Kentucky, at the end of quarters 2 and 6 of the vaccination campaign.

	% with	% with	% with		Average
Quarter of Vaccination	High School Degree	Bachelor Degree	Graduate Degree	% Employed	Family Income
Q2: 31 May 2021	−0.83	0.90	0.85	0.50	0.75
Q6: 31 May 2022	−0.73	0.80	0.74	0.50	0.67

Notes: Education measures are reported for the population 25 and older. Employment and health insurance measures are reported for the population 16 and older.

## Data Availability

The datasets analyzed in this study cannot be publicly accessible.

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
