# Peer review of "Geographical Patterns of COVID-19 Vaccine Inequality by Race and Ethnicity and Sociodemographic Determinants of Health: Evidence from Louisville, Kentucky"

_vaccines, 2025, doi:10.3390/vaccines13121241_

Round 1
Reviewer 1 Report
Comments and Suggestions for Authors
Please review my word doc review-
Manuscript ID: Vaccines 2025, 13, x
Title: Geographical Patterns of COVID-19 Vaccine Inequality by Race and Ethnicity and Sociodemographic Determinants of Health: Evidence from Louisville, Kentucky
Journal: Vaccines (MDPI)
Recommendation: Accept with Revisions
General Comments:
I have personally reviewed the manuscript titled "Geographical Patterns of COVID-19 Vaccine Inequality by Race and Ethnicity and Sociodemographic Determinants of Health: Evidence from Louisville, Kentucky."
This study addresses a persistent and critical issue in public health which is the intersection of race, geography, and social determinants of health (SDOH) in vaccine uptake1. The authors utilize ZIP code-level data from the Kentucky Immunization Registry to map disparities in Louisville, KY, over a two-year period2.
Overall, I find the manuscript to be methodologically sound and scientifically relevant. While the broad confirmation that racial minorities and lower-income populations have lower vaccination rates is not new333, the value of this paper lies in its granularity.
By drilling down to the ZIP code level and overlaying SDOH variables, the authors move beyond the "ecological fallacy" often found in county-level studies4. This is an important issue as the logical error of drawing conclusions about individuals based on aggregate data from the group they belong to. It assumes that what is true for a group is automatically true for every individual within that group, which can lead to misleading interpretations because individual-level relationships may differ from group-level ones They effectively demonstrate that race functions largely as a proxy for socioeconomic factors like education and income within this specific metropolitan context5.
The paper is very readable, well-structured, the data sources are authoritative (registry data vs. survey data) 6, and the visualizations 7777 clearly communicate the spatial nature of these disparities. I recommend publication after the authors address a few minor points regarding data limitations and the discussion of causality.
Specific Comments for the Authors
- Novelty and Granularity
There are many studies related to challenges within cohorts and populations to understand the motivations for vaccine acceptance. The introduction frames the study well by acknowledging that while racial disparities are known, spatial heterogeneity at the neighborhood level is often overlooked8. I appreciated the finding that Black residents in predominantly White/affluent ZIP codes had higher vaccination rates than those in predominantly Black/lower-income ZIP codes9. This is a crucial nuance that challenges monolithic views of racial vaccine hesitancy. It would be beneficial if the discussion emphasized this point even more strongly as a key takeaway for designing local interventions.
- Methodology and Data Handling
- Imputation of Demographics: You mention using an algorithm to address missing demographic data and the over-reporting of "Multiracial" and "Some Other Race" categories10. While you cite a prequel article for the full methodology, a brief sentence or two in the Methods section summarizing how this algorithm works (e.g., did it rely on census ratios?) would save the reader from having to hunt down the external reference.
- Ecological Inference: The analysis correlates aggregate ZIP code statistics (income/education) with vaccination rates11. The manuscript is generally careful about this, but please ensure the Limitations section explicitly notes that individual-level behavior cannot be perfectly inferred from these aggregate metrics. You touch on this12, but reinforcing that high-income individuals in low-income ZIP codes (or vice versa) act as outliers would strengthen the statistical honesty.
- Results and Visualization
- Figures 1 & 2: These maps are excellent and tell the story of the "East/West divide" in Louisville very clearly13131313.
- Data Consistency: In Table 2, the shift from Q2 to Q6 shows significant growth14. It is interesting that the "catch-up" rate was high in the West region, yet the disparity remained. The text handles this interpretation well.
- Discussion of Causality vs. Access
The paper establishes a strong correlation between SDOH (specifically education) and vaccination rates15. However, the distinction between access barriers (e.g., clinic hours, transportation) and hesitancy (trust in the system) is somewhat blended. Since this is a registry-based study, you cannot definitively measure hesitancy. I suggest softening the language in the conclusion to reflect that while SDOH correlates with uptake, the specific mechanism (access vs. trust) remains a hypothesis to be tested by qualitative follow-up.
- Minor Corrections
- Please review the capitalization of "Black" and "White" to ensure consistency throughout the text, as per standard academic style guides (e.g., in the Introduction, "white individual" is lowercase16, but capitalized elsewhere).
- Check the formatting of the citation numbers in the introduction to ensure they align with the journal's style guide.
Conclusion
This is a solid contribution to the literature on vaccine equity. It provides actionable evidence for local health departments in Kentucky and serves as a replicable model for other metropolitan areas conducting spatial analysis of health outcomes. I look forward to seeing the final version.
Author Response
Open Review
( ) I would not like to sign my review report
(x) I would like to sign my review report
Quality of English Language
( ) The English could be improved to more clearly express the research.
(x) The English is fine and does not require any improvement.
|
Yes |
Can be improved |
Must be improved |
Not applicable |
|
|
Does the introduction provide sufficient background and include all relevant references? |
(x) |
( ) |
( ) |
( ) |
|
Is the research design appropriate? |
(x) |
( ) |
( ) |
( ) |
|
Are the methods adequately described? |
(x) |
( ) |
( ) |
( ) |
|
Are the results clearly presented? |
(x) |
( ) |
( ) |
( ) |
|
Are the conclusions supported by the results? |
(x) |
( ) |
( ) |
( ) |
|
Are all figures and tables clear and well-presented? |
(x) |
( ) |
( ) |
( ) |
Comments and Suggestions for Authors
General Comments
I have personally reviewed the manuscript titled "Geographical Patterns of COVID-19 Vaccine Inequality by Race and Ethnicity and Sociodemographic Determinants of Health: Evidence from Louisville, Kentucky."
This study addresses a persistent and critical issue in public health which is the intersection of race, geography, and social determinants of health (SDOH) in vaccine uptake1. The authors utilize ZIP code-level data from the Kentucky Immunization Registry to map disparities in Louisville, KY, over a two-year period2.
Overall, I find the manuscript to be methodologically sound and scientifically relevant. While the broad confirmation that racial minorities and lower-income populations have lower vaccination rates is not new333, the value of this paper lies in its granularity.
By drilling down to the ZIP code level and overlaying SDOH variables, the authors move beyond the "ecological fallacy" often found in county-level studies4. This is an important issue as the logical error of drawing conclusions about individuals based on aggregate data from the group they belong to. It assumes that what is true for a group is automatically true for every individual within that group, which can lead to misleading interpretations because individual-level relationships may differ from group-level ones They effectively demonstrate that race functions largely as a proxy for socioeconomic factors like education and income within this specific metropolitan context5.
The paper is very readable, well-structured, the data sources are authoritative (registry data vs. survey data) 6, and the visualizations 7777 clearly communicate the spatial nature of these disparities. I recommend publication after the authors address a few minor points regarding data limitations and the discussion of causality.
Specific Comments for the Authors
- Novelty and Granularity
- There are many studies related to challenges within cohorts and populations to understand the motivations for vaccine acceptance. The introduction frames the study well by acknowledging that while racial disparities are known, spatial heterogeneity at the neighborhood level is often overlooked8. I appreciated the finding that Black residents in predominantly White/affluent ZIP codes had higher vaccination rates than those in predominantly Black/lower-income ZIP codes9. This is a crucial nuance that challenges monolithic views of racial vaccine hesitancy. It would be beneficial if the discussion emphasized this point even more strongly as a key takeaway for designing local interventions.
Authors’ response: You summarized it beautifully. Thank you! We borrowed your language and added a couple of sentences at the end of the first paragraph of the discussion section.
- Methodology and Data Handling
- Imputation of Demographics:You mention using an algorithm to address missing demographic data and the over-reporting of "Multiracial" and "Some Other Race" categories10. While you cite a prequel article for the full methodology, a brief sentence or two in the Methods section summarizing how this algorithm works (e.g., did it rely on census ratios?) would save the reader from having to hunt down the external reference.
Authors’ response: Thank you for the comment. We have revised the entire limitation paragraph in the discussion section and added more details about the algorithm.
- Ecological Inference:The analysis correlates aggregate ZIP code statistics (income/education) with vaccination rates11. The manuscript is generally careful about this, but please ensure the Limitations section explicitly notes that individual-level behavior cannot be perfectly inferred from these aggregate metrics. You touch on this12, but reinforcing that high-income individuals in low-income ZIP codes (or vice versa) act as outliers would strengthen the statistical honesty.
Authors’ response: This is an important point. Thank you for emphasizing that. We have now expanded our note on the matter in the discussion section's last paragraph. Our updated note (we borrowed from your language) now reads as:
“Moreover, individual-level behavior cannot be perfectly inferred from the findings of this study, which was conducted at the ZIP code level and thus overlooks individual-level variation within ZIP codes, as residents in the same ZIP code might not share the same socioeconomic conditions or access to vaccines. For example, high-income individuals in low-income ZIP codes (or vice versa) act as outliers.”
- Results and Visualization
- Figures 1 & 2: These maps are excellent and tell the story of the "East/West divide" in Louisville very clearly13131313.
Authors’ response: Thank you. We appreciate it.
- Data Consistency: In Table 2, the shift from Q2 to Q6 shows significant growth14. It is interesting that the "catch-up" rate was high in the West region, yet the disparity remained. The text handles this interpretation well.
Authors’ response: Thank you. We appreciate it.
- Discussion of Causality vs. Access
- The paper establishes a strong correlation between SDOH (specifically education) and vaccination rates15. However, the distinction between access barriers (e.g., clinic hours, transportation) and hesitancy (trust in the system) is somewhat blended. Since this is a registry-based study, you cannot definitively measure hesitancy. I suggest softening the language in the conclusion to reflect that while SDOH correlates with uptake, the specific mechanism (access vs. trust) remains a hypothesis to be tested by qualitative follow-up.
Authors’ response: Thank you for the thoughtful comment. We borrowed from your language and revised the paragraph in the discussion section. Particularly, the last sentence of the paragraph now reads as:
“The precise mechanism—whether it is access or trust—is still a hypothesis that requires testing through further qualitative studies. Hence, strategies to improve racial and ethnic SDOH disparities may need to be combined with policies to strengthen vaccine acceptance by increasing trust in the healthcare system, communicating vaccine efficacy information effectively to historically abused communities, and engaging trusted community stakeholders with extensive networks.”
- Minor Corrections
- Please review the capitalization of "Black" and "White" to ensure consistency throughout the text, as per standard academic style guides (e.g., in the Introduction, "white individual" is lowercase16, but capitalized elsewhere).
Authors’ response: Thank you for catching the inconsistency. We found them in the text and corrected them.
- Check the formatting of the citation numbers in the introduction to ensure they align with the journal's style guide.
Authors’ response: We have used the journal’s template in preparing the draft. What we used seemed to be what the journal recommends, but are certainly open to changing the referencing style however the journal requests.
Conclusion
This is a solid contribution to the literature on vaccine equity. It provides actionable evidence for local health departments in Kentucky and serves as a replicable model for other metropolitan areas conducting spatial analysis of health outcomes. I look forward to seeing the final version.
Submission Date
12 November 2025
Date of this review
27 Nov 2025 18:52:48
Reviewer 2 Report
Comments and Suggestions for Authors
The core of article is well written. As currently written, it ignores multiple relevant dimensions of COVID-19 infections, infections post vaccinations, COVID-19 vaccine adverse events like myocarditis (relevant to vaccine hesitancy), length of protection of full COVID-19 vaccinations before breakthrough infections start occurring (note: no mention of breakthrough infections), etc. The authors are encouraged to address these deficiencies.
Lines 55-57 - may be an overestimate - please replace with studies comparing outcomes between COVID-19 vaccinated and unvaccinated groups with timeframes of at least 12 months.
Line 74 - geography is likely the wrong term for this proxy - consider neighborhood or equivalent.
Line 113 - Please cite URL of data source.
Line 133 - Please consider releasing the Python programs as open source (github.com or equivalent).
Table 1 - "Frequencies" line immediately below the header line does not make sense to this reviewer - the data appears to be population counts not frequencies.
Table 1 Notes: "3.3. Formatting of Mathematical Components." needs explanation - this reviewer does not understand the information being presented.
Table 2 - Consider widening column 8 at the expense of column 1 (or the blank column spacers).
Please add a description of the correlation calculations used in Table 4, etc. to the methods section.
Table 4 - Are these Pearson correlation coefficients or some other coefficients?
Lines 304-305 - Relaxing social distancing concept doesn't really work when breakthrough infections are occuring for fully vaccinated and also boosted COVID-19 vaccinees.
Comments on the Quality of English Language
The English is well written.
Author Response
Open Review
( ) I would not like to sign my review report
(x) I would like to sign my review report
Quality of English Language
(x) The English could be improved to more clearly express the research.
( ) The English is fine and does not require any improvement.
Authors’ Response: We read the manuscript a few more times, corrected grammatical errors, revised many sentences for clarity, and reformatted the figures and tables.
|
Yes |
Can be improved |
Must be improved |
Not applicable |
|
|
Does the introduction provide sufficient background and include all relevant references? |
( ) |
(x) |
( ) |
( ) |
|
Is the research design appropriate? |
(x) |
( ) |
( ) |
( ) |
|
Are the methods adequately described? |
( ) |
(x) |
( ) |
( ) |
|
Are the results clearly presented? |
( ) |
(x) |
( ) |
( ) |
|
Are the conclusions supported by the results? |
(x) |
( ) |
( ) |
( ) |
|
Are all figures and tables clear and well-presented? |
( ) |
(x) |
( ) |
( ) |
Comments and Suggestions for Authors
The core of article is well written. As currently written, it ignores multiple relevant dimensions of COVID-19 infections, infections post vaccinations, COVID-19 vaccine adverse events like myocarditis (relevant to vaccine hesitancy), length of protection of full COVID-19 vaccinations before breakthrough infections start occurring (note: no mention of breakthrough infections), etc. The authors are encouraged to address these deficiencies.
Authors’ Response: We have responded to this comment in conjunction with the next comment.
Lines 55-57 - may be an overestimate - please replace with studies comparing outcomes between COVID-19 vaccinated and unvaccinated groups with timeframes of at least 12 months.
Authors’ Response: Thank you very much for these very thoughtful comments. We spent a good time thinking about them. We ultimately decided (1) to delete the sentence in lines 55-57 to avoid such hyperbolic statements and (2) not to dig deep into such matters, as the article is merely about assessing geographical differences in vaccination rate by race and ethnicity and their correlations with SDOH. We do not attempt to describe the clinical and epidemiological characteristics of COVID-19 or render any judgment on the effectiveness of COVID-19 vaccines. Nonetheless, we accounted for the breakthrough infections when responding to the comments on lines 304-305.
Line 74 - geography is likely the wrong term for this proxy - consider neighborhood or equivalent.
Authors’ Response: We revised the sentence to: “The neighborhood of residence is another key SDOH proxy”
Line 113 - Please cite URL of data source.
Authors’ Response: We have provided the link now. Thank you.
Line 133 - Please consider releasing the Python programs as open source (github.com or equivalent).
Authors’ Response: We have released all the Python programs used in this analysis on GitHub. They can be found at https://github.com/Hamidz88/Geographical_Inequality_COVID-19_Vaccine_Race_Ethnicity/tree/main.
Table 1 - “Frequencies” line immediately below the header line does not make sense to this reviewer - the data appears to be population counts not frequencies.
Authors’ Response: We understand it. We replaced the terms “Frequencies” and “Percentages” with “Population” and “Shares”.
Table 1 Notes: “3.3. Formatting of Mathematical Components.” needs explanation - this reviewer does not understand the information being presented.
Authors’ Response: That’s a typo. Thank you for catching it. We deleted it.
Table 2 - Consider widening column 8 at the expense of column 1 (or the blank column spacers).
Authors’ Response: We took care of that. We also edited the entire table to look better
Please add a description of the correlation calculations used in Table 4, etc. to the methods section.
Authors’ Response: Certainly! We revised the following sentence in the first paragraph of the Materials and Methods section
“Additionally, the associations between ZIP code vaccination rates and a set of SDOH (namely, the ZIP code-level rates of educational attainment, employment status, insurance status, and average family income) were assessed.”
to read:
“Additionally, correlations between ZIP code-level vaccination rates and a set of SDOH (namely, educational attainment, employment status, insurance status, and average family income) were calculated using the Pearson product-moment correlation coefficient.”
Table 4 - Are these Pearson correlation coefficients or some other coefficients?
Authors’ Response: Yes, we mentioned that in our response to the previous comment.
Lines 304-305 - Relaxing social distancing concept doesn’t really work when breakthrough infections are occuring for fully vaccinated and also boosted COVID-19 vaccinees.
Authors’ Response: That is a great point. Thank you very much. We have now accounted for that by revising the sentence to read as:
“Nonetheless, workplace behaviors and vaccine acceptance share a bidirectional relation-ships when fully vaccinated individuals are not experiencing breakthrough infections: a higher need for interpersonal relationships increases vaccination uptake, and vaccination enables the relaxation of social distancing protocols.”
Comments on the Quality of English Language
The English is well written.
Submission Date
12 November 2025
Date of this review
03 Dec 2025 22:54:14